# The epidemiology of Mayaro virus in the Americas: A systematic review and key parameter estimates for outbreak modelling

Edgar-Yaset Caicedo[1], Kelly Charniga[2], Amanecer Rueda[1], Ilaria Dorigatti[2], Yardany Mendez[1], Arran Hamlet[2], Jean-Paul Carrera[3,4], Zulma M. Cucunubá[2]*

**1** Universidad Pedagógica y Tecnológica de Colombia, Tunja, Colombia, **2** MRC Centre for Global Infectious Disease Analysis (MRC-GIDA), Imperial College London, London, United Kingdom, **3** Department of Zoology, University of Oxford, Oxford, United Kingdom, **4** Department of Research in Virology and Biotechnology, Gorgas Memorial Institute of Health Studies, Panama City, Panama

☯ These authors contributed equally to this work.
* zulma.cucunuba@imperial.ac.uk

**Data Availability Statement:** All the data associated to this paper is available at https://github.com/zmcucunuba/mayv-review.

## Abstract

Mayaro virus (MAYV) is an arbovirus that is endemic to tropical forests in Central and South America, particularly within the Amazon basin. In recent years, concern has increased regarding MAYV's ability to invade urban areas and cause epidemics across the region. We conducted a systematic literature review to characterise the evolutionary history of MAYV, its transmission potential, and exposure patterns to the virus. We analysed data from the literature on MAYV infection to produce estimates of key epidemiological parameters, including the generation time and the basic reproduction number, $R_0$. We also estimated the force-of-infection (FOI) in epidemic and endemic settings. Seventy-six publications met our inclusion criteria. Evidence of MAYV infection in humans, animals, or vectors was reported in 14 Latin American countries. Nine countries reported evidence of acute infection in humans confirmed by viral isolation or reverse transcription-PCR (RT-PCR). We identified at least five MAYV outbreaks. Seroprevalence from population based cross-sectional studies ranged from 21% to 72%. The estimated mean generation time of MAYV was 15.2 days (95% CrI: 11.7–19.8) with a standard deviation of 6.3 days (95% CrI: 4.2–9.5). The per-capita risk of MAYV infection (FOI) ranged between 0.01 and 0.05 per year. The mean $R_0$ estimates ranged between 2.1 and 2.9 in the Amazon basin areas and between 1.1 and 1.3 in the regions outside of the Amazon basin. Although MAYV has been identified in urban vectors, there is not yet evidence of sustained urban transmission. MAYV's enzootic cycle could become established in forested areas within cities similar to yellow fever virus.

## Author summary

Each year, diseases that are transmitted by mosquitoes cause substantial deaths and disability across the world. We performed a systematic literature review of Mayaro virus (MAYV) and estimated key epidemiological parameters that can be used to improve

**Funding:** EYC is funded by a young researcher grant (Joven Investigador) UPTC, Colciencias call No. 018 2018. KC is funded by Imperial College London's President's PhD Scholarship. ID acknowledges research funding from a Sir Henry Dale fellowship funded by Wellcome Trust and The Royal Society (grant 213494/Z/18/Z). JPC is funded by the Clarendon Scholarship from University of Oxford and Lincoln-Kingsgate Scholarship from Lincoln College, University of Oxford (grant number SFF1920_CB2_MPLS_1293647). ZMC is funded by the MRC/Rutherford Fund Research Fund (grant MR/R024855/1). ZMC, ID and KC jointly acknowledge the MRC Centre for Global Infectious Disease Analysis (reference MR/R015600/1), jointly funded by the UK Medical Research Council (MRC) and the UK Foreign, Commonwealth & Development Office (FCDO), under the MRC/FCDO Concordat agreement and is also part of the EDCTP2 programme supported by the European Union. The funders played no role in the study design, data collection and analysis, decision to publish, or preparation of the manuscript.

**Competing interests:** The authors have declared that no competing interests exist.

future outbreak response. We estimated the generation time and basic reproduction number for a historical outbreak. Our results suggest that the force-of-infection of MAYV in endemic settings is low. We did not find evidence of substantial urban transmission of MAYV. Nevertheless, similarities between MAYV and yellow fever virus epidemiology suggest that MAYV could emerge in urban areas. Local transmission of MAYV has never been reported outside of Central and South America. Our results highlight the need to continue monitoring emerging arboviruses in the Americas.

## Introduction

Mayaro virus (MAYV) is an enveloped, single-stranded RNA virus with a complex transmission cycle involving mosquitoes and animals, including non-human primates, birds, horses, rodents, and reptiles [1–3]. MAYV belongs to the *Togaviridae* family and *Alphavirus* genus. Along with Una virus (UNAV), it is classified as a new world member of the Semliki forest antigenic complex [4]. Phylogenetic studies have identified a least three MAYV genotypes, D (widely dispersed), L (limited), and N (new) [5–7], with limited geographic distribution possibly linked to host range and vector habitat suitability. In 1954, MAYV was first discovered in forest workers in Mayaro County, Trinidad and Tobago [8]. Since then, the virus has caused sporadic outbreaks of febrile disease in Central and South America [5,9–12].

Recent research suggests that MAYV is spreading in the Americas, with autochthonous cases of MAYV reported in Venezuela in 2010 [5] and in Haiti in 2014 and 2015 [13,14]. MAYV-dengue co-circulation was also identified in Brazil in 2011–2012 [15]. In 2020, French health authorities reported 13 laboratory-confirmed cases of Mayaro fever in French Guiana over a period of only three months. About one to three confirmed cases were reported each year from 2017–2019. Notably, 11 of these cases lived in urban areas but travel histories were not available at the time of writing [16]. These examples highlight the public health importance of MAYV as an emerging pathogen.

In humans, clinically acute MAYV infections are characterized by a febrile disease. The most common signs and symptoms are fever and headache. Myalgia, eye pain, chills, arthralgia, rash, and cough are less frequently reported. Longitudinal studies have shown that some patients continue to experience joint pain up to one year after infection in a similar way to those infected by chikungunya virus [17]. Due to nonspecific symptoms, cases of Mayaro fever may resemble dengue fever cases or cases of other tropical diseases, such as malaria. This makes clinical diagnosis of MAYV a challenge in regions where multiple arboviruses circulate simultaneously. Past infection can also be difficult to ascertain; MAYV exhibits serological cross-reactivity with other alphaviruses. This means that infection with one alphavirus, such as chikungunya, may lead to a rise in MAYV antibodies, even in those who have never been infected by MAYV [18]. There is currently no approved vaccine or specific treatment for MAYV infection, though at least three vaccine candidates have been developed [19–21].

Although several reviews of MAYV epidemiology and transmission have been published recently [2,22–26], key epidemiological parameters, such as the generation time and the basic reproduction number, have not been estimated from MAYV data. Consequently, there is a dearth of mathematical modelling studies of MAYV [18,27]. By conducting a systematic review of the literature and estimating key parameters, we aim to fill important knowledge gaps on MAYV in order to understand its transmission dynamics in the Americas. The results of this analysis can be used to anticipate future spread and disease burden, which can improve outbreak preparedness and guide public health interventions.

## Methods

### Systematic review

A systematic literature review (up to January 11, 2019) was undertaken to collate data on MAYV transmission, exposure, and phylogenetics in humans, animals, and vectors. We searched all peer-reviewed publications and grey literature in PubMed, Web of Science, Literatura Latino-Americana e do Caribe em Ciencias da Saúde (LILACS), Google Scholar, and Excerpta Medica database (EMBASE) for publications containing the terms *Mayaro* or *Uruma*. Though once considered distinct viruses, Uruma is now considered a strain of MAYV [2]. Further details regarding search terms can be found in S1 Text. We did not restrict by date of publication or language. For our analysis, we included publications with information on: (i) the time of exposure to MAYV; (ii) the time of symptom onset; (iii) viral load data; (iv) susceptibility of mosquitoes to MAYV; (v) times series of outbreaks in humans; (vi) age-stratified seroprevalence data. Each article was allocated to two reviewers who independently screened abstracts and titles. We excluded articles that: (i) were not about MAYV; (ii) were not in English, Spanish, Portuguese, or French. There was only one paper that was not written in the aforementioned languages; it was written in Chinese and was excluded due to lack of fluency among the authors. Then, two reviewers independently reviewed the full text of each article. We contacted authors to obtain additional information as needed. Disagreements were resolved by a third reviewer. The systematic review was conducted in agreement with the Preferred Reporting Items for Systematic Review and Meta-analyses guidelines (S1 PRISMA Checklist) [28].

Data extracted from publications were classified into three categories: humans, animals, and vectors. For humans, data were further classified into four sub-categories: (i) case reports, (ii) outbreaks, (iii) hospital-based surveillance, and (iv) cross-sectional seroprevalence surveys. Animals were further classified according to taxonomic order (S1 Text). Data extracted from the literature are available on GitHub at https://github.com/zmcucunuba/mayv-review.

### Force-of-infection models

We used catalytic models fitted to age-stratified seroprevalence data to estimate the historical per-capita risk of infection [29]. Denoting $n(a,t)$ the number of seropositive individuals of age $a$ at time $t$, $N$ the serosurvey sample size, and $P(a,t)$ the underlying seroprevalence at age $a$ at time $t$, we assumed that the number of seropositive subjects follows a Binomial distribution $n(a,t) \sim B(N,P(a,t))$. For force-of-infection (FOI) that is constant over time, denoted $\lambda$, we modelled seroprevalence for age $a$ in year $t$ (i.e. the time when the serosurvey occurred) as $P(a,t) = 1 - \exp(-\lambda a)$. For time-varying FOI ($\lambda_t$), seroprevalence for age $a$ was given by $P(a, t) = 1 - \exp(-\sum_{i=t-a+1}^{t} \lambda_i)$.

Model assumptions included no loss of antibodies over time, no differences in susceptibility or exposure by age, and no differences in the mortality rate of infected versus susceptible individuals. Models were implemented in Stan using the No-U-Turn sampler, a type of Hamiltonian Monte Carlo sampling. We ran 10,000 iterations with a burn-in period of 5,000. After convergence, the best-fitting models were selected based on the expected log predictive density for an out-of-sample data point (elpd) [30,31].

### Generation time and instantaneous reproduction number

The generation time is the time between sequential rounds of infection, specifically between infection of a human case and infection of the secondary human cases caused by the primary case. The generation time has been previously estimated for other arboviruses, including Zika virus [32], chikungunya virus [33], and dengue virus [34] but not MAYV. We estimated the

generation time for MAYV using data from travellers to endemic regions, human viral clearance data, mosquito mortality rate, and experimental vector studies. Parameter inference was conducted in a Bayesian framework using Markov chain Monte Carlo (MCMC) methods. Further details can be found in S1 Text.

The instantaneous reproduction number $R_t$ is the average number of secondary cases generated by a primary case over the course of the infectious period. This value measures transmissibility, which is important for planning and modifying public health responses [35]. We estimated $R_t$ for MAYV using the R package EpiEstim [36] for the 1954–1955 outbreak in Santa Cruz, Boliva for which the incidence (number of reported cases by week of onset) was available. Further details are provided in the S1 Text.

## Phylogenetic analysis

Phylogenetic reconstruction of MAYV was carried out in order to describe the geographical distribution and genotypes. For this, sixty-five complete genome sequences were obtained from the GenBank library and aligned and manually edited using the MUSCLE algorithm implemented in MEGA7 software and Seaview software [37,38]. The best-fitting nucleotide substitution model was selected with jModelTest 2 software [39,40] based on Bayesian information criterion (BIC). Lower values of BIC are preferred, and a difference of about 6 is considered meaningful [41]. The maximum-likelihood tree was constructed by generalised time-reversible + invariable sites + gamma model with IQ-TREE software. The statistical robustness of the tree topology was evaluated with Ultrafast bootstrap support and Shimodaira-Hasegawa-like approximate likelihood-ratio test (SH-aLRT) with 2,000 replicates [42,43]. Further details are provided in the S1 Text.

## Results

We included 76 publications that describe MAYV transmission: five outbreaks, 19 case reports, 13 hospital-surveillance studies, 11 seroprevalence studies, 17 studies with possible evidence of MAYV, five studies in vectors, and 13 studies in animals. Some publications fell into more than one category, and further details can be found in S1 Text.

### Evidence of MAYV in human populations

**Case reports.** We found 19 case reports that describe between one and 13 MAYV infections in Central and South America (Fig 1 and S1 Text). All infections were confirmed by viral isolation, presence of IgM antibodies, or PCR. A total of 61 individual cases were reported; of these, eight were reported in travellers from the United States, France, Germany, Switzerland, and Netherlands. All travellers visited areas within the Amazon basin area (S1 Text) and none of them transmitted MAYV to secondary cases. All except five cases [8] were detected since 1995.

**Outbreaks.** We found evidence of seven potential MAYV outbreaks in the literature. Five studies provided clear evidence of MAYV outbreaks in Brazil, Bolivia, and Venezuela (Figs 1 and 2 and Table 1). We were unable to find the original sources that described the 1991 outbreaks in Para and Tocantins states in Brazil [4]. Mosquitoes infected with MAYV were found in only two out of the five documented outbreaks. In both of these outbreaks, the identified species was *Haemagogus janthinomys*.

**Hospital-based surveillance.** Thirteen studies described hospital-based surveillance for MAYV in febrile patients. A combination of IgM and PCR methods were used to confirm infection. The proportion of confirmed MAYV in febrile patients ranged between 0% and 20% (Fig 3 and S1 Text).

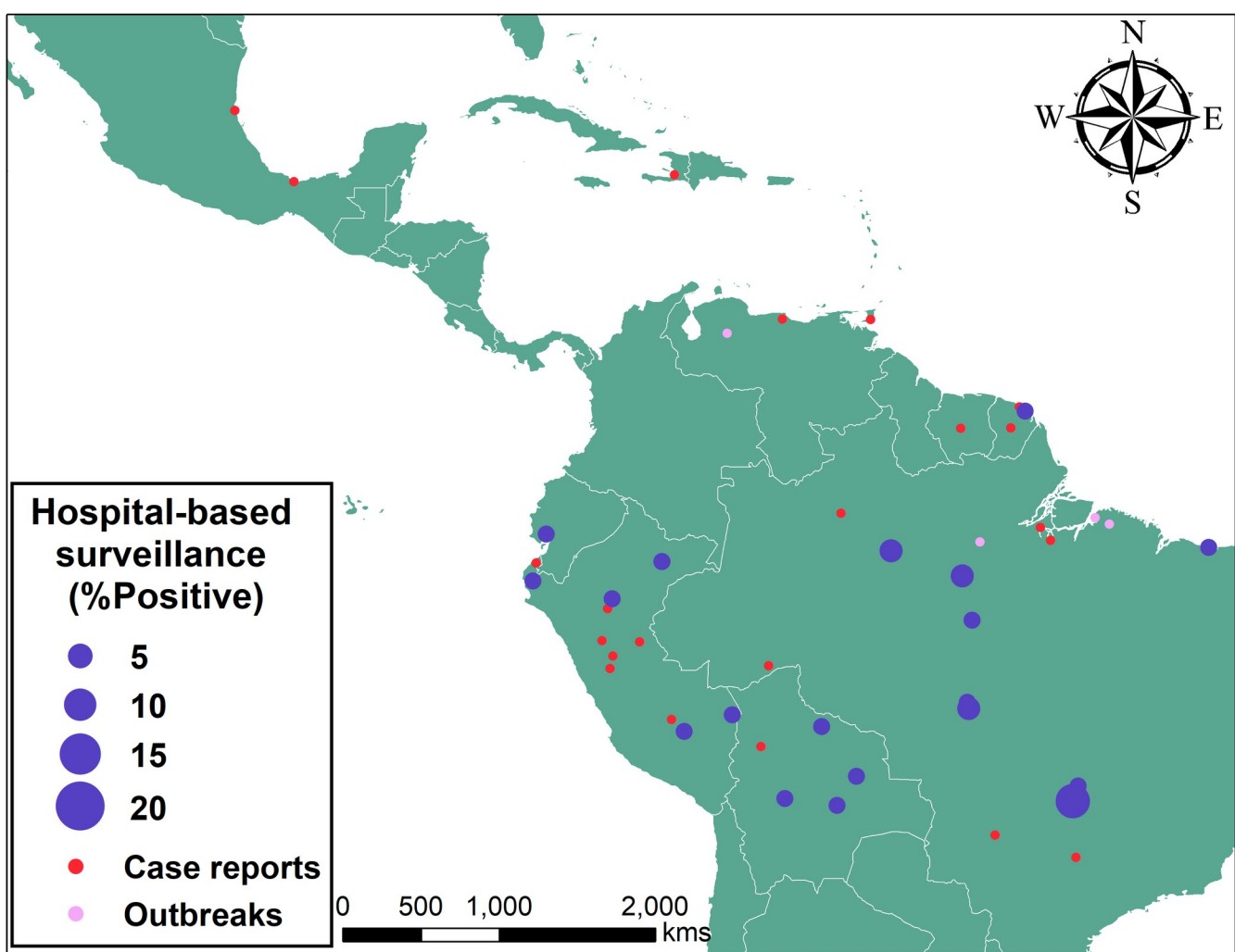

**Fig 1. Map of confirmed cases of MAYV from case reports, outbreaks, and hospital-based surveillance.** Map was created with ArcGIS Desktop 10.6 using shapefiles from Esri. Data sources for the shapefiles include Esri, Garmin International Inc., US Central Intelligence Agency, and National Geographic Society [44].

**Cross-sectional seroprevalence studies.** Eleven cross-sectional seroprevalence studies were identified that met the inclusion criteria. The prevalence of IgG/IgM antibodies against MAYV is presented in Fig 3 and S1 Text. The range of seroprevalence for all cross-sectional studies was 2%-68%. For population-based studies, the range was 6%-67%.

**Other possible evidence of MAYV transmission.** Seventeen studies demonstrated possible evidence of MAYV transmission in humans (S1 Text). These studies had less diagnostic certainty than the studies in other categories, but still strongly suggest MAYV presence. Most of these studies considered other arboviruses in addition to MAYV and involved native populations.

## Evidence of MAYV in animals

Thirteen studies described evidence of MAYV transmission in animals. Of those studies, only one isolated the virus from a bird of the species *Icterus spurius* in Louisiana, United States in 1967 [46]. Another study in Brazil used reverse transcription-PCR (RT-PCR) to test primates

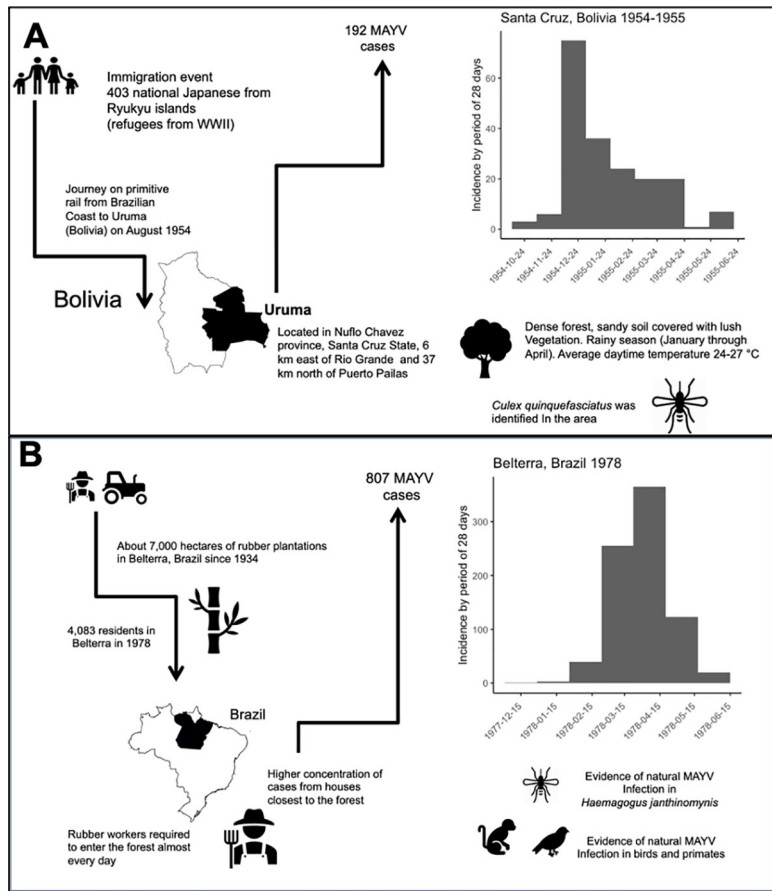

**Fig 2.** MAYV outbreaks in A) Santa Cruz, Bolivia in 1954–1955 and B) Belterra, Brazil in 1978.

for viral RNA targeting the genera *Flavivirus* and *Alphavirus*. All primates tested negative for current infection; however, two different species, *Sapajus xanthosternos* and *Ateles marginatus*, tested positive by PRNT for MAYV antibodies indicative of past exposure [47]. In addition to this study, eleven other studies tested animals for MAYV antibodies. Antibodies were reported in several different types of animals, especially primates. The proportion of animals that tested positive for MAYV antibodies across studies ranged from 0.4%-100% in primates (mean 44.6%). Among all other animals, the proportion that tested positive ranged from 0.4%-60% with mean 13.1%. The proportion of animals that tested positive for MAYV by order and family is shown in Fig 4. Further details about the animal studies are reported in S1 Text.

## Evidence of MAYV in mosquitoes

Evidence of MAYV in eight species of sylvatic mosquitoes was described in seven studies (Table 2). However, only *Haemagogus janthinomys* has tested positive in the context of human outbreaks [9,48]. The first known cases of MAYV in humans in Trinidad and Tobago were associated with *Manzonia venezuelensis* [49]. Positive pools of *Psorophora fexox* and *Ps. Albipes* were found in Colombia in the 1950s [50] and in Panama in the 1960s, along with *Culex voremifer* [51]. Notably, pools of *Aedes aegypti*, an important vector for Zika, dengue, and chikungunya virus transmission, tested positive for MAYV in Mato Grosso, Brazil in 2013 [52]. *Ae. aegypti* is a competent vector of MAYV under laboratory conditions [53] and has been found

**Table 1. Characteristics of MAYV outbreaks (n = 5).**

| Ref | Years | Town/state/country | Confirmatory tests[*] | Number of confirmed/total suspected/discarded cases | Attack rate | Phylogenetic | Observations | Zone | Population type | Detection in vectors | Detection in animals |
|---|---|---|---|---|---|---|---|---|---|---|---|
| [12] | 1954–1955 | Uruma, Santa Cruz, Bolivia | Culture + NT | 3/192/none | 47.6%[**] | Isolate later classified as Genotype D [9] | Yellow fever virus and Ilheus virus antibodies in asymptomatic people | rural (50–75% forest cover) | Migrants (Ryukyu Islands, Japan) | No evidence of *Aedes* but *Culex* found in the area. | not reported |
| [45] | 1955 | Sao Miguel do Guama, Para, Brazil | Culture + NT and HI | 16/91/75 | 18.7% | Isolate later classified as Genotype D [9] | Simultaneous malaria outbreak | rural (75–100% forest cover) | Locals that travelled to forest | no | not reported |
| [10] | 1977–1978 | Belterra, Para, Brazil | Seroconversion by HI | 55/807/none | 19.4%[***] | Isolates from vectors and humans from Para in the 1970s identified as Genotype D and L [4,5] | Simultaneous yellow fever outbreak | rural (75–100% forest cover) | Locals from urban area who work in the forest | *Haemagogus janthinomys* | Birds: *Columbiformes*, *Caprimulgiformes*, *Passeriformes* Primates: *Cebidae*, *Callithricidae*, *Callithrix* |
| [9] | 2008 | Pau D'arco, Para, Brazil | IgM ELISA/culture | 36/105/69 | | Genotype D | | rural (75–100% forest cover) | Temporary visitors (agronomy students from Belem) | *Haemagogus janthinomys* | not reported |
| [5] | 2010 | Ospino, Portuguesa, Venezuela | Culture | 6/77/none | | Genotype D | | rural/peri-urban (75–100% forest cover) | Locals (65% women) | no | not reported |

[*]NT: neutralization test, HI: hemagglutination inhibition.

[**]Attack rate was calculated for suspected cases only divided by the total population.

[***]Serological survey in 1972 found a seroprevalence of 10.3% in this population. In July 1978, the serosurvey was repeated in the same population and a seroprevalence of 29.7% was reported. The attack rate was estimated by the difference between these two surveys.

naturally infected in parks and gardens within urban areas [52]. Research has also shown that *Ae. albopictus*, another important human disease vector, is able to transmit MAYV to mice in the laboratory [54].

## Epidemiological parameters

**MAYV generation time.** We estimated an overall mean MAYV generation time of 15.2 days (95% credible interval [CrI]: 11.7–19.8) with standard deviation 6.2 days (95% CrI: 4.2–9.5). The components of the generation time distribution can be found in Table 3.

**MAYV force-of-infection.** We identified 11 age-stratified seroprevalence studies that used either enzyme-linked immunosorbent assay (ELISA), plaque reduction neutralization test (PRNT), neutralization test (NT), or hemmaglutination inhibition (HI) techniques to quantify the antibody levels against MAYV. We used these data to estimate the force-of-

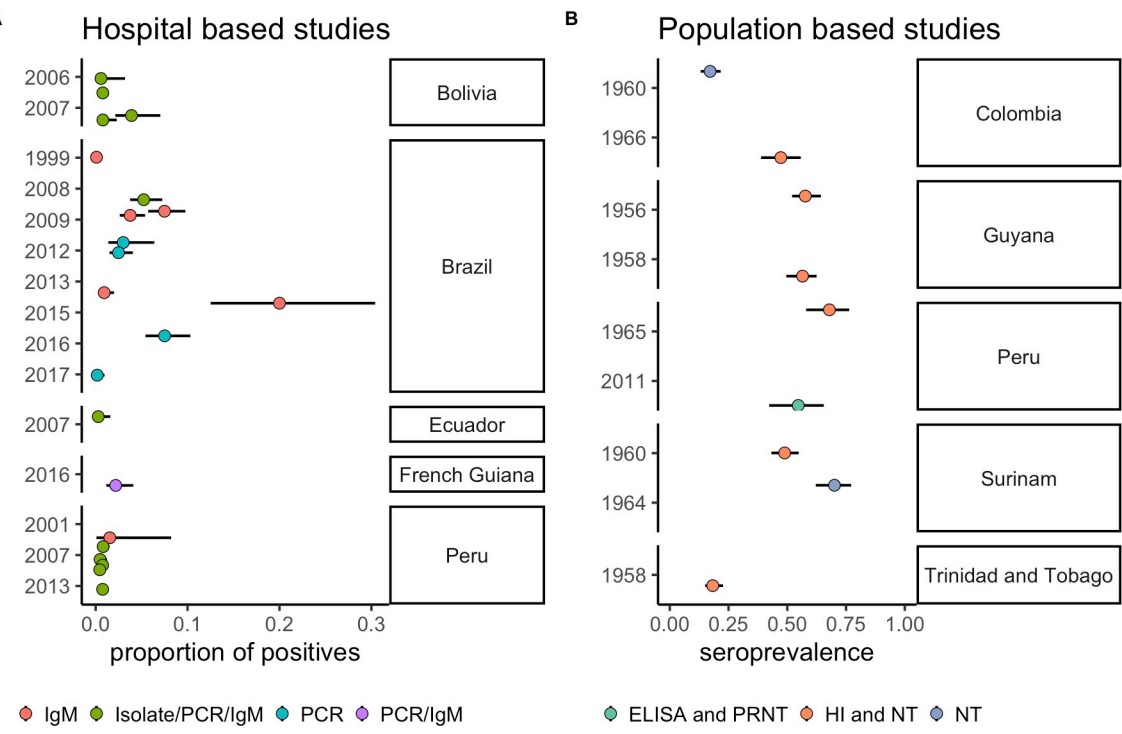

**Fig 3. Hospital-based surveillance and cross-sectional seroprevalence studies of MAYV.** Points indicate the mean and lines represent the 95% confidence interval. Colours indicate the type of test used to confirm cases.

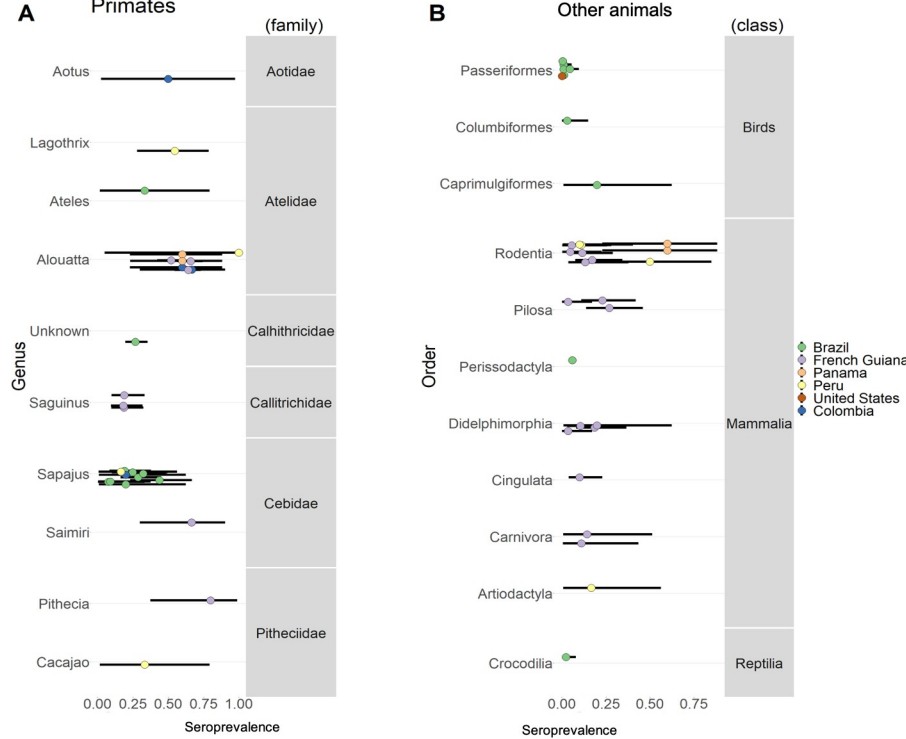

**Fig 4. Proportion of animals that tested positive for MAYV antibodies across studies.** A) Primates and their corresponding taxonomic family and genus. B) Other animals and their corresponding taxonomic class and order. Colours indicate study location. Points indicate the mean and lines represent the 95% confidence interval.

**Table 2. Evidence of natural infection of MAYV in mosquitoes.**

| Ref | Species | Country | State | Town/city | Isolation year | Diagnostic method* | Number of Pools | Number of Positive pools | Proportion of positive pools | Zone |
|---|---|---|---|---|---|---|---|---|---|---|
| [49] | *Mansonia venezuelensis* | Trinidad and Tobago | | | 1957 | CF, M-HI, direct NT | | 1 | | Primary forest/ rural area |
| [55] | *Psorophora albipes/ferox* | Colombia | Santander | San Vicente de Chucuri | 1958 | Direct NT, ST, M-HI, CF [56] | 276 | 4 | 0.1% | Forests and cultured fields in rural area |
| [51] | *Psorophora ferox* | Panama | Bocas del Toro | Almirante | 1961 | M-HI and CF [56] | 93 | 1 | 1.1% | Forest close to the town |
| [51] | *Culex voremifer* | Panama | Bocas del Toro | Almirante | 1966 | 1 hamster infected (M-HI) | 153 | 1 | 0.6% | Forest close to the town |
| [11] | *Haemagogus janthinomys* | Brazil | Para | Belterra | 1978 | M-HI [56] | 62 | 9 | 14.5% | Forest/rubber plantations close to houses |
| [9] | *Haemagogus janthinomys* | Brazil | Para | Santa Barbara | 2008 | C6/36 cells culture, M-CF, immunofluorescence assays | 11 | 1 | 9.1% | Peri-domiciliary in rural area |
| [52] | *Aedes aegypti* | Brazil | Mato Grosso | Cuiaba | 2013 | RT-PCR, Culture (Vero cells, C6/36 cells) | 171 | 4 | 2.3% | urban |
| [52] | *Culex spp.* | Brazil | Mato Grosso | Cuiaba | 2013 | RT-PCR, Culture (Vero cells, C6/36 cells) | 403 | 12 | 2.9% | urban |
| [57] | *Sabethes spp* | Brazil | Para | Belem | 1965 | M-HI [56] | 48 | 2 | 4.1% | |

*ST: serological test with guinea pigs, rhesus monkeys, and mice serum. M-H: mouse hemagglutination M-HI: mouse hemagglutination inhibition M-CF: complement fixation, H: hemagglutination, HI: hemagglutination inhibition, NT: neutralization test, RT-PCR: reverse transcription polymerase chain reaction.

infection (FOI) and population-level exposure patterns to MAYV for seven countries (Fig 5). Reconstruction of the historical FOI from seroprevalence studies in the Americas suggest an almost constant FOI, or endemic transmission, in the Amazon basin countries of Colombia, Guyana, Peru, and Suriname. Only two studies outside of the Amazon basin, in Colombia and Trinidad and Tobago, suggest a time-varying FOI, a pattern that is consistent with epidemic transmission.

**MAYV $R_0$ and $R_t$.** Seroprevalence studies were used to estimate $R_0$, the basic reproduction number. $R_0$ estimates ranged from an average of 1.11 (95% CrI: 1.09–1.12) in 1960 in French Guiana to an average of 3.47 (95% CrI: 1.53–7.92) in 1966 in Brazil (Table 4). We estimated a mean $R_0$ between 2.1 and 2.9 in the Amazon basin areas and between 1.1 and 1.3 in the regions outside of the Amazon basin.

We used the weekly incidence of Mayaro fever cases reported in Santa Cruz, Bolivia in 1954–1955 to estimate the time-varying reproduction number, $R_t$ (Fig 6). The first estimate of

**Table 3. Estimates of the mean and standard deviation of the generation time distribution and its components.**

| | Mean (95% CrI) | Standard deviation (95% CrI) | Source |
|---|---|---|---|
| **Intrinsic incubation period (days)** | 3.0 (2.2–3.8) | 1.2 (1.0–1.7) | estimated |
| **Time to viral clearance (days)** | 3.9 (3.5–4.4) | 1.0 (0.8–1.2) | estimated |
| **Human-to-mosquito generation time (days)** | 3.4 (2.0–4.8) | 0.7 (0.5–1.1) | estimated |
| **Extrinsic incubation period (days)** | 9.4 (8.4–10.7) | 4.6 (3.3–6.7) | estimated |
| **Mosquito life time (days) (as for *Aedes aegypti*)** | 5.3 (fixed) | 1.4 (fixed) | [32] |
| **Mosquito-to-human generation time (days)** | 11.9 (8.6–16.3) | 6.2 (4.2–9.5) | estimated |
| **Mayaro virus generation time (days)** | 15.2 (11.7–19.8) | 6.2 (4.2–9.5) | estimated |

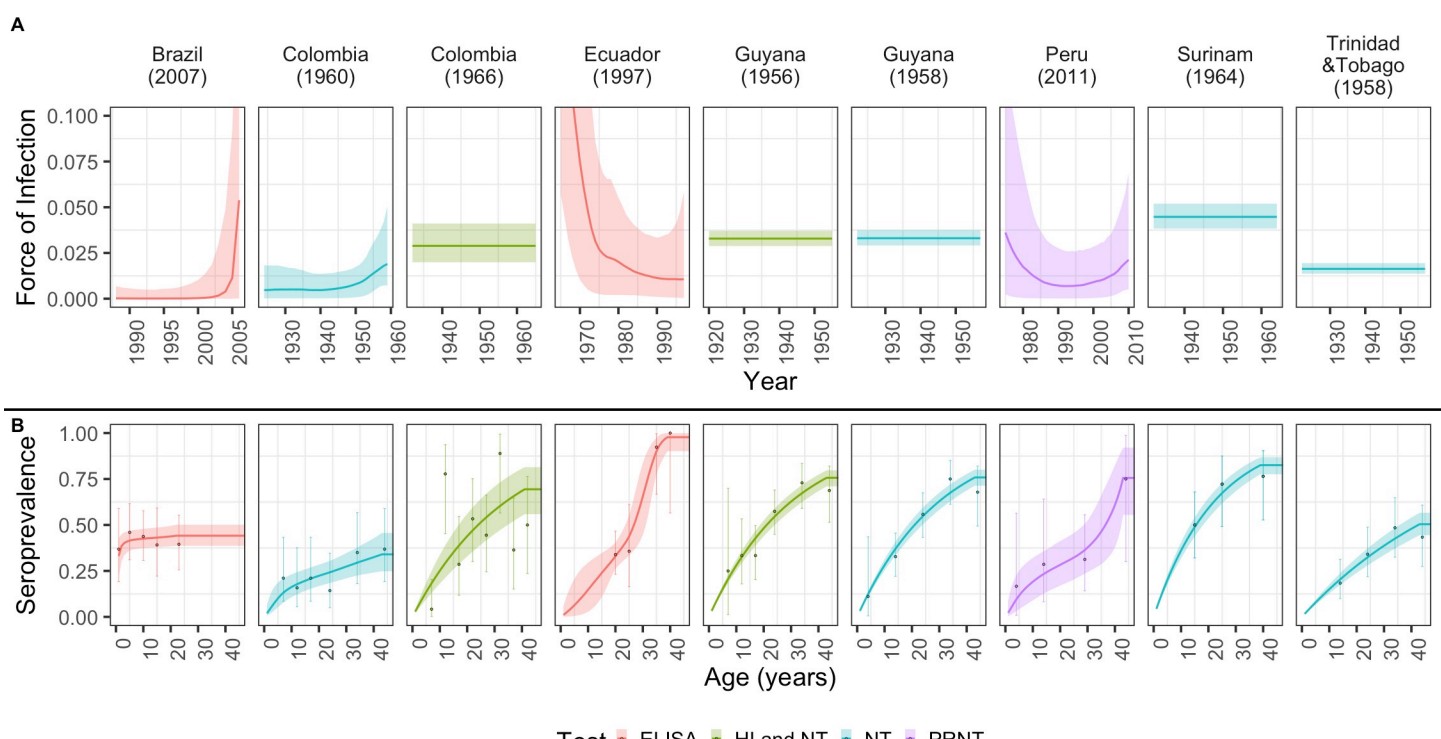

**Fig 5. MAYV exposure patterns in Latin America.** (A) FOI estimates obtained from catalytic models fitted to MAYV age-stratified seroprevalence data. B) Data (points) and estimated (line) age-stratified seroprevalence against MAYV. Points and dark lines represent mean estimates; bars and shaded areas represent the 95% CrI. Colours indicate the type of test used to confirm cases.

**Table 4. Basic reproduction number estimates obtained from seroprevalence studies of MAYV.**

| Ref | Country | Study year | Test* | $R_0$ Mean (95% CrI) |
|---|---|---|---|---|
| [58] | Brazil | 1965 | HI | 1.21 (1.01–1.98) |
| [58] | Brazil | 1966 | HI | 3.47 (1.53–7.92) |
| [58] | Brazil | 1970 | HI | 1.30 (1.03–2.13) |
| [58] | Brazil | 1972 | HI | 1.86 (1.19–3.94) |
| [58] | Brazil | 1972 | HI | 1.78 (1.18–3.63) |
| [59] | Brazil | 2007 | ELISA | 1.15 (1.00–2.20) |
| [50] | Colombia | 1960 | NT | 1.23 (1.04–1.66) |
| [60] | Colombia | 1966 | HI and NT | 2.10 (1.71–2.69) |
| [61] | Ecuador | 1997 | ELISA | 2.99 (1.30–8.10) |
| [62] | French Guiana | 1996 | HI | 1.11 (1.09–1.12) |
| [63] | Guyana | 1956 | HI and NT | 2.27 (2.09–2.48) |
| [64] | Guyana | 1958 | NT | 2.29 (2.11–2.51) |
| [65] | Peru | 2011 | PRNT | 1.57 (1.04–3.59) |
| [66] | Suriname | 1964 | NT | 2.92 (2.59–3.31) |
| [64] | Trinidad and Tobago | 1958 | NT | 1.58 (1.47–1.70) |

*HI: hemagglutination inhibition, NT: neutralization test, PRNT: plaque reduction neutralization test, ELISA: enzyme-linked immunosorbent assay.

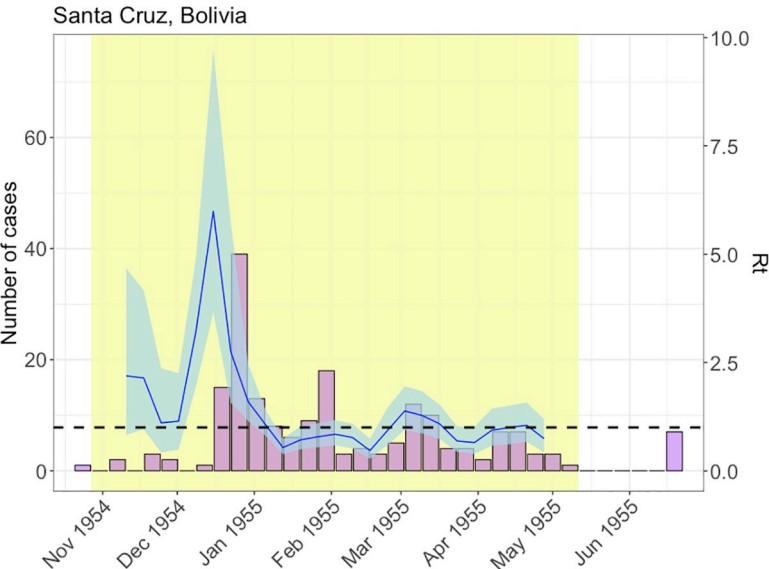

**Fig 6. Time-varying reproduction number estimates for the 1954–1955 MAYV outbreak in Santa Cruz, Bolivia.**
Mayaro fever cases (left axis and bars) overlaid with estimates of the reproduction number, *R*, over time (right axis, running 4-week average shown, centred on the middle week). Blue shaded areas show 95% CrI around the median estimated *R*. Yellow shaded areas show the part of the time series used to estimate *R*. The threshold value *R* = 1 is indicated by the horizontal dashed line.

$R_t$ in a susceptible population is equivalent to $R_0$. We estimated a mean $R_0$ at the beginning of the outbreak of 2.2 (95% CrI: 0.8–4.8).

**Phylogenetic analysis.** We found 59 whole-genome sequences of MAYV from humans (53 sequences) and mosquitoes (six sequences) comprising three genotypes: L, N, and D (Fig 7).

Genotype L was first identified in Pará, Brazil in 1955. Since then, it has been found in other locations within the Brazilian Amazon. This genotype has only been reported outside of the Brazilian Amazon three times; in 2013 it was identified in *Ae. aegypti* in Mato Grosso, Brazil [52], and in 2014 [14] and 2015 [13], it was isolated from two human cases in Haiti. Genotype N has not been detected outside of Peru. The only known isolate came from a human in the department of Madre de Dios in the Amazon basin [5]. Genotype D was first reported in Trinidad and Tobago in 1955. Between 1955 and 2014, it was identified in several countries in the Amazon jungle (Brazil, Bolivia, Peru, and French Guiana) [4,5]. In 2010, it was identified for the first time outside of the Amazon jungle in the state of Portuguesa, in the North of Venezuela [5] and in 2014, it was detected in Haiti [13]. Genotype D has been reported in five MAYV outbreaks (Table 1).

## Discussion

Based on the earliest date from our FOI models, MAYV has been circulating for at least 90 years in the Amazon basin and other forested areas in Central and South America. Although recent reports recognise MAYV as a public health threat due to its epidemic potential, little is known about its epidemiology and burden in endemic regions. Our findings suggest that autochthonous transmission of MAYV occurs in humans, vectors, or animals in at least 14 countries in the Americas. The most important contribution of our study is the estimation of key epidemiological parameters, including the incubation period in humans, the generation

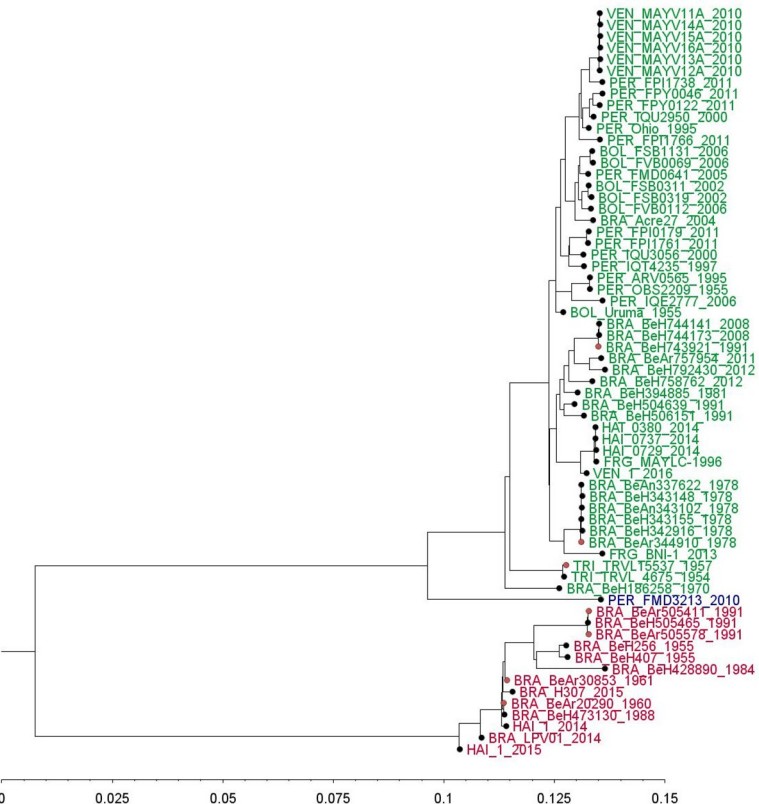

**Fig 7. Genetic relationships between MAYV strains by midpoint-rooted maximum-likelihood.** Three genotypes of MAYV are shown: L (limited, in red), N (new, in blue), and D (widely dispersed, in green). Origin from isolates: mosquitoes (pink dots) and humans (black dots). The scale is the percentage of divergence from the origin.

time, and the basic reproduction number. These parameters characterise the natural history of MAYV infection and can be used to improve disease prevention and surveillance strategies.

Although non-human primates are thought to be the principle enzootic reservoir of MAYV, evidence of previous infection with MAYV has been found in a wide range of animals [46,48]. High seroprevalence has been documented in several types of mammals, such as primates and rodents, with lower seroprevalence reported in birds and reptiles. The 1978 MAYV outbreak in Belterra, Brazil is the only outbreak where MAYV antibodies were detected in animals in the surrounding areas, including in the *Columbiformes*, *Caprimulgiformes*, and *Passeriformes* families of birds and in the *Cebidae* and *Callithricidae* families of primates [10,48]. We identified only one study in our systematic review that reported viral isolation of MAYV in animals, a bird in Louisiana [46]. This study cited a reference that did not come up in our review, which listed MAYV isolations from two species of lizard, *Ameiva ameiva* and *Tropidurus hispidus*, in Utinga Forest, Pará, Brazil in the early 1960s [67]. However, according to the 1965 Belém Virus Laboratory Annual Report, the two isolations from lizards were "technically suspect" and there had been "no isolations [of MAYV] from wild vertebrates" in the Brazilian Amazon [68]. The importance of animals in human transmission of MAYV is still unclear.

At least eight species of mosquitoes can be naturally infected with MAYV, but only *Haemagogus janthinomys* has been associated with MAYV outbreaks in humans [9,48]. In 2013, MAYV-infected *Ae. aegypti* mosquitoes were found in urban areas in Brazil [52]. This species

of mosquito has been implicated in large epidemics of other arboviruses in Latin America in recent years, including Zika and dengue. More widespread infection of *Ae. aegypti* with MAYV could have important ramifications for viral emergence and spread. Genetic changes in arboviruses could increase their ability to infect other vectors, potentially shifting the transmission cycle from sylvatic to urban [25]. A single mutation in the chikungunya virus has been attributed to increasing the virus' infectivity for *Ae. albopictus*, leading to more efficient viral dissemination in the mosquitoes and transmission to mice in the laboratory [69]. The mutation has allowed chikungunya virus to spread into new areas with low levels of population immunity [70]. However, genetic mutations favouring such adaptations have not yet been detected in MAYV [25].

Our estimate of the generation time (mean 15.2 [95% CrI: 11.7–19.8] days with standard deviation 6.2 [95% CrI: 4.2–9.5] days) for MAYV is comparable to that of chikungunya virus. For chikungunya virus, Salje et al. estimated a mean generation time of 14.0 days with standard deviation of 6.2 days [33]. Although the generation time distribution here was estimated independently of temperature, the component parameters that depend on the vector can vary based on temperature. For example, as temperature increases within a range acceptable to the vector, dengue virus replicates faster, decreasing the extrinsic incubation period (EIP) of *Ae. aegypti* [71]. Additionally, mosquito longevity depends on temperature and species [72]. It is unclear which species of mosquito is most important in MAYV transmission. Moreover, due to limited data, we had to pool information on six different species of mosquitoes to estimate the EIP. Thus, in order to estimate a temperature-dependent generation time distribution for MAYV, more vector data is needed.

Our estimates of the basic reproduction number $R_0$ from seroprevalence studies (range 1.1–3.5) and from a time series of cases from an outbreak in Bolivia (2.2 [95% CrI: 0.8–4.8]) are also similar to estimates for other arboviruses. For example, a review study reported an average $R_0$ of 4.25, 2.98, and 3.09 for dengue, Zika, and chikungunya viruses, respectively, in tropical regions [73]. Another study estimated the $R_0$ for MAYV using parameters fitted to data from a 2018 chikungunya virus outbreak in Rio de Janeiro. They obtained estimates ranging from 1.18 to 3.51 based on the assumption that both viruses can infect and potentially be transmitted by the same species of mosquitoes [27].

In the literature, higher seroprevalence of MAYV has been reported in cross-sectional studies compared to hospital-based surveillance (mean 39% and mean 3% across all studies, respectively). Typically, higher seroprevalence is expected in hospital-based surveillance because the patients already have symptoms. One reason that could explain lower seroprevalence in hospital-based studies is if those studies were carried out in the midst of other arbovirus outbreaks. In fact, all but two studies tested for other arboviruses.

One of the most intriguing findings from our review concerns evidence of locally transmitted MAYV infection in Haiti from two studies. Blohm et al. reported five human isolates from Haiti in 2014, which were classified as Genotype L and D [6]. Previously, Mavian et al. suggested that an isolate from Haiti in 2015 corresponded to Genotype L and may have been the result of a recombination event originating from a São Paulo strain and Acre ancestors, which came from Mato Grosso [7]. These studies represent the first reports of Mayaro fever cases in Haiti and could indicate regional spread of the virus.

Future spread of MAYV may also be affected by cross-protective immunity in which serological cross-reactivity confers immunity to infection with other viruses. Cross-protective immunity between alphaviruses may limit MAYV spread and emergence potential. A recent study found that mice that were exposed to chikungunya virus and a chikungunya virus vaccine in the laboratory exhibited strong and moderate cross-protection, respectively, when challenged with MAYV [74]. They also found that human sera from patients that had been

infected by chikungunya virus cross-neutralised MAYV at high titres. Based on their findings *in vitro*, the authors suggest that herd immunity to chikungunya, which is high in Latin America following the 2013–2015 epidemics, may limit the spread of MAYV in that region [74].

Further evidence of cross-protective immunity among alphaviruses comes from experimental studies on hamsters. One study showed that hamsters immunized with Venezuelan equine encephalitis vaccines experienced a reduction in mortality of 37% and 59% after challenge with Western equine encephalitis and Eastern equine encephalitis viruses, respectively [75]. Whether cross-protective immunity among alphaviruses is observed in human populations *in vivo* is unknown. Population-based studies are currently underway to evaluate the effects of cross-protective immunity on alphavirus epidemiology and emergence.

Though genetically unrelated, MAYV shares numerous similarities with yellow fever virus (YFV) in South America. YFV is a flavivirus that affects both humans and non-human primates (NHPs). In South America, almost all human cases of yellow fever are due to sylvatic spillover mediated by mosquitoes of the genera *Haemagogus* and *Sabethes* [76]. In rare instances, YFV can establish itself in the absence of the NHP reservoir and human-to-human transmission via *Ae. aegypti* can occur. Historically, YFV has been mostly confined to the Amazon basin. However, the virus has rapidly expanded outside of its endemic zone over the last 20 years. The largest YFV outbreaks since the 1940s started in Brazil's southeastern states in 2016 [76]. Over two thousand human yellow fever cases have been reported over this time period, including at least 749 deaths [77], and transmission is ongoing as of 2020 [78]. The persistence of YFV across multiple transmission seasons in southeastern Brazil suggests that endemicity may have been achieved.

MAYV and YFV appear to primarily affect those who enter forested regions for economic activity, and several NHP species can be infected by both viruses [79]. These similarities in geographic spread and epidemiology should not be overlooked. If MAYV continues its southward spread through Mato Grosso do Sul and Sao Paulo states, as YFV did in the early 2000s, then we may see a similar pattern of large-scale epidemics in southern Brazil [80]. Given MAYV's ability to infect *Ae. aegypti*, high vector densities in cities across the continent, and the large scale of historical urban yellow fever outbreaks in South America [81], further urban spread could be devastating. Lessons learnt from the wide body of research on YFV could be used to guide research on MAYV.

A limitation of this review is lower levels of diagnostic certainty for some of the studies. Due to cross-reactivity among alphaviruses [18], some Mayaro fever cases could have been misclassified as diseases resulting from infection by other alphaviruses. The highest level of diagnostic certainty (gold standard) is achieved with viral culture. We did not exclude studies that used less certain diagnostic methods. Consequently, the MAYV $R_0$ from seroprevalence studies may have been overestimated due to serological cross-reactivity among other circulating alphaviruses. In contrast, cross-reactivity likely did not affect the estimation of $R_0$ from the MAYV outbreak in Bolivia. The individuals affected by the outbreak were migrants from Japan and would have had few opportunities to be infected by other alphaviruses. Although asymptomatic individuals tested positive for yellow fever virus and Ilheus virus antibodies, these viruses are flaviviruses and therefore are not expected to affect the immune response to an alphavirus. The studies included in this review also likely suffer from underreporting of MAYV incidence due to overlap of symptoms with other arboviral diseases, including chikungunya and dengue. Also, there is uncertainty and risk of bias due to the small number of Mayaro fever cases included in the intrinsic incubation period analysis. Another limitation is that the use of cross-sectional seroprevalence studies did not allow us to investigate changes in the FOI over time or differences in exposure by age. One assumption of the catalytic models applied in our analysis is that there are no differences in exposure by age, which seems

plausible when assuming no previous exposure such as in Brazil in 2017 (Fig 5B). However, we acknowledge that in some situations, adults may have had higher exposure to MAYV than children due to working in forested areas. Unfortunately, we do not have access to such data.

The results of this study highlight evidence of MAYV circulation in the Americas dating back nearly a century. Enzootic transmission cycles of this virus involve *Haemagogus* mosquitoes and non-human primates, commonalities which are shared by YFV. Taken together, our findings suggest that adaptive mutations of the virus and invasion of forested areas within cities could trigger MAYV emergence and spread.

## Supporting information

**S1 PRISMA Checklist. Preferred Reporting Items for Systematic Reviews and Meta-Analyses.**
(DOC)

**S1 Text. Fig A. Flowchart showing the selection of studies. Fig B. Viral load detected in plasma in Mayaro infected cases per day post symptoms onset**. Mean and range for 21 patients are shown. **Fig C. Maximum likelihood EIP probability density function (left) and cumulative distribution function (right)**. The aggregated proportion of mosquitoes that tested positive at the relative days post-infection are shown as bars. **Table A. Boolean algorithms for literature search. Table B. Data classification of MAYV studies in humans. Table C. Values used in estimate_R() function in EpiEstim package. Table D. Characteristics of Mayaro fever case reports. Table E. Characteristics of Mayaro fever cases included in the intrinsic incubation period analysis (N = 15). Table F. Characteristics of hospital-based surveillance studies included in the analysis. Table G. Characteristics of MAYV cross-sectional seroprevalence studies. Table H. Studies with possible evidence of MAYV transmission**. These studies were not classified in other categories but strongly indicate presence of MAYV. **Table I. Studies that detected MAYV in animals. Table J. Full genomes of MAYV included in the phylogenetic analysis. Table K. Nucleotide substitution models**. The best-fitting model is in bold.
(DOCX)

## Acknowledgments

We would like to acknowledge Alberto Cumbrera from Gorgas Memorial Institute, Panama for his help with Fig 1.

## Author Contributions

**Conceptualization:** Edgar-Yaset Caicedo, Kelly Charniga, Yardany Mendez, Jean-Paul Carrera, Zulma M. Cucunubá.

**Data curation:** Edgar-Yaset Caicedo, Kelly Charniga, Amanecer Rueda, Yardany Mendez.

**Formal analysis:** Edgar-Yaset Caicedo, Kelly Charniga, Ilaria Dorigatti, Zulma M. Cucunubá.

**Funding acquisition:** Zulma M. Cucunubá.

**Investigation:** Edgar-Yaset Caicedo, Kelly Charniga, Amanecer Rueda.

**Methodology:** Kelly Charniga, Ilaria Dorigatti, Zulma M. Cucunubá.

**Project administration:** Zulma M. Cucunubá.

**Software:** Ilaria Dorigatti, Zulma M. Cucunubá.

**Supervision:** Jean-Paul Carrera, Zulma M. Cucunubá.

**Validation:** Kelly Charniga, Amanecer Rueda.

**Visualization:** Edgar-Yaset Caicedo, Kelly Charniga, Ilaria Dorigatti, Zulma M. Cucunubá.

**Writing – original draft:** Edgar-Yaset Caicedo, Kelly Charniga, Ilaria Dorigatti, Yardany Mendez, Arran Hamlet, Jean-Paul Carrera, Zulma M. Cucunubá.

**Writing – review & editing:** Kelly Charniga, Ilaria Dorigatti, Yardany Mendez, Arran Hamlet, Jean-Paul Carrera, Zulma M. Cucunubá.

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
