## [Decision Letter · Decision Letter 0]

16 Jan 2021

Dear Dr Cucunuba,

Thank you very much for submitting your manuscript "The epidemiology of Mayaro virus in the Americas:  A systematic review and key parameter estimates for outbreak modelling" for consideration at PLOS Neglected Tropical Diseases. As with all papers reviewed by the journal, your manuscript was reviewed by members of the editorial board and by several independent reviewers. The reviewers appreciated the attention to an important topic. Based on the reviews, we are likely to accept this manuscript for publication, providing that you modify the manuscript according to the review recommendations. 

Sincerely,

Eugenia Corrales-Aguilar

Deputy Editor

Eugenia Corrales-Aguilar

Deputy Editor

Reviewer's Responses to Questions

**Key Review Criteria Required for Acceptance?**

**Methods**

-Are the objectives of the study clearly articulated with a clear testable hypothesis stated?

-Is the study design appropriate to address the stated objectives?

-Is the population clearly described and appropriate for the hypothesis being tested?

-Is the sample size sufficient to ensure adequate power to address the hypothesis being tested?

-Were correct statistical analysis used to support conclusions?

-Are there concerns about ethical or regulatory requirements being met?

Reviewer #1: My two minor comments are:

1/ The choice to exclude studies not in English, French, Spanish or Portuguese requires justification. This may not be all that difficult as I'd think most if not all important studies would be available in these languages, but nonetheless scientific (as above) and/or pragmatic (language fluency among the authors) justification should be presented.

2/ The meaning of the statement that "Each study was allocated to two reviewers who independently screened abstracts and titles" (lines 107-8), most specifically the meaning of "study" in this context. I'd infer that this means the areas covered in the paper ("(i) the time of exposure to MAYV; (ii) the time of symptom onset....") but the statement could be misconstrued as written.

Reviewer #2: I am really satisfied with the exposition of the methods and the application of the statistical models. The methods and models described in the paper were clearly exposed and relevant for this study. The authors used all available information in the literature to derive the epidemiological parameters. See attached review for more details.

Reviewer #3: (No Response)

**Results**

-Does the analysis presented match the analysis plan?

-Are the results clearly and completely presented?

-Are the figures (Tables, Images) of sufficient quality for clarity?

Reviewer #1: Again, minor comments:

1/ Figure 3 needs slight editing as some of the data points (coloured circles) have been clipped at the bottom or top.

2/ The text above Table 3 includes all (or perhaps nearly all) the results from the table. These results should be in the table or text, but not both (at least not in full).

3/ What are "Ileus antibodies" (Table 1). I'm not sure if I should know, but I don't. If this is not widely known an explanation should be given (or if it's a typo it should be corrected).

Reviewer #2: The paper clearly presents 1/ how papers were selected, 2/ the different types of MAYV transmission (case reports, outbreaks, MAYV in animals, etc..) 3/ How the epidemiological parameters can be derived from these data.

Reviewer #3: (No Response)

**Conclusions**

-Are the conclusions supported by the data presented?

-Are the limitations of analysis clearly described?

-Do the authors discuss how these data can be helpful to advance our understanding of the topic under study?

-Is public health relevance addressed?

Reviewer #1: The conclusions are well supported by the materials presented.

Reviewer #2: The discussion covers the limitations of the study.

Reviewer #3: (No Response)

**Editorial and Data Presentation Modifications?**

Reviewer #1: (No Response)

Reviewer #2: I recommend publication of the manuscript after some minor revisions:

1. Introduction Lines 62-64. The sentence on serological cross-reactivity seems out of place here. I would recommend addressing this point around line 85 where the authors discuss the difficulty of diagnosis. 

2. Page 16 Table 3. The parameters are assumed to be the same for all studies. I was wondering if we could expect variations of the epidemiological parameters, for instance with the climatic conditions. (Mordecai Plos NTD 2013) discuss the impact of temperature on arbovirus transmission. Can we expect the mosquito lifetime to depend on the temperature and the season? If so, how would it change the different parameters?

3. Figure 5 Page 17. The authors applied serocatalytic models to assess the force of infection (FOI) from age stratified seroprevalence studies. They tested models of time-independent (endemicity) and time-dependent FOI. However, other models could explain the data, and from cross-sectional seroprevalence studies it is not possible to distinguish variations in time of the FOI or different exposures with age. For instance, the curve from Ecuador 1997 could also be due to an outbreak that happened in the 1990s and where adults were more exposed than children. It would be good to discuss this possibility, even more so that MAYV is often considered, as yellow fever, to be more likely to infect adults that are active near forested areas.

Reviewer #3: (No Response)

**Summary and General Comments**

Reviewer #1: This is an ambitious paper that presents a lot of interesting information. Other than my comments above the paper is well written and presented. I wonder if you could update the literature search beyond 11 January 2019?

Reviewer #2: (No Response)

Reviewer #3: In this work, the authors wrote a review on the Mayaro virus circulation in the Americas. Using the reported data, they estimate epidemiological parameters such as the basic reproduction number for different outbreaks.

The manuscript is well written and helps to shed the light on the importance of surveillance data to keep track of emergent arboviruses. In the case of the Mayaro virus, they report the presence of the MAYV for around 100 years in the Americas. The methodology protocols are well explained which allows the reader to follow the data gathering and analysis. While I consider this work to be an important contribution, there are some minor adjustments to the text and figures that are necessary and some questions that need to be clarified:

Minor suggestions:

Details of the protocols are provided in the main text. Even though the used model is widely used by the community, details about the applied equations and model per se could be included in the Support Information.

The authors used several different software for performing the simulation and for building the phylogenetic tree. It would be useful for the community if the codes and scripts used in these calculations to be provided in the SI. Also, the data extracted from the literature and organized could also be made available for reproducibility.

I suggest increasing the font size of all Figures’ text to help the reader.

The authors should expand the discussion about the possible overestimation of the Mayaro R0 due to the serological cross-reactivity with other viruses.

PLOS authors have the option to publish the peer review history of their article (what does this mean?). If published, this will include your full peer review and any attached files.

Reviewer #1: Yes: David Harley

Reviewer #2: No

Reviewer #3: Yes: Vinícius G. Contessoto
---

## [Decision Letter · Decision Letter 1]

27 Apr 2021

Dear Dr Cucunuba,

We are pleased to inform you that your manuscript 'The epidemiology of Mayaro virus in the Americas:  A systematic review and key parameter estimates for outbreak modelling' has been provisionally accepted for publication in PLOS Neglected Tropical Diseases.

Best regards,

Eugenia Corrales-Aguilar

Deputy Editor

Eugenia Corrales-Aguilar

Deputy Editor

Reviewer's Responses to Questions

**Key Review Criteria Required for Acceptance?**

**Methods**

-Are the objectives of the study clearly articulated with a clear testable hypothesis stated?

-Is the study design appropriate to address the stated objectives?

-Is the population clearly described and appropriate for the hypothesis being tested?

-Is the sample size sufficient to ensure adequate power to address the hypothesis being tested?

-Were correct statistical analysis used to support conclusions?

-Are there concerns about ethical or regulatory requirements being met?

Reviewer #1: (No Response)

Reviewer #2: Yes

Reviewer #3: (No Response)

**Results**

-Does the analysis presented match the analysis plan?

-Are the results clearly and completely presented?

-Are the figures (Tables, Images) of sufficient quality for clarity?

Reviewer #1: (No Response)

Reviewer #2: Yes

Reviewer #3: (No Response)

**Conclusions**

-Are the conclusions supported by the data presented?

-Are the limitations of analysis clearly described?

-Do the authors discuss how these data can be helpful to advance our understanding of the topic under study?

-Is public health relevance addressed?

Reviewer #1: (No Response)

Reviewer #2: Yes

Reviewer #3: (No Response)

**Editorial and Data Presentation Modifications?**

Reviewer #1: (No Response)

Reviewer #2: (No Response)

Reviewer #3: (No Response)

**Summary and General Comments**

Reviewer #1: Dear Authors

Thank you for revising your manuscript. I am now satisfied with your responses to my comments and those from the other reviewers.

Reviewer #2: The authors addressed all points raised by myself and other reviewers. I recommend publication of their article.

Reviewer #3: The authors addressed all my questions and suggestions. I recommend the publication in the current version of the manuscript.

PLOS authors have the option to publish the peer review history of their article (what does this mean?). If published, this will include your full peer review and any attached files.

Reviewer #1: **Yes: **David Harley

Reviewer #2: No

Reviewer #3: **Yes: **Vinícius G Contessoto

---

## [Editor Report · Acceptance letter]

28 May 2021

Dear Dr Cucunuba,

We are delighted to inform you that your manuscript, "The epidemiology of Mayaro virus in the Americas:  A systematic review and key parameter estimates for outbreak modelling," has been formally accepted for publication in PLOS Neglected Tropical Diseases.

Best regards,

Shaden Kamhawi

co-Editor-in-Chief

Paul Brindley

co-Editor-in-Chief
